# The Effect of Sr-CoFe_2_O_4_ Nanoparticles with Different Particles Sized as Additives in CIP-Based Magnetorheological Fluid

**DOI:** 10.3390/ma14133684

**Published:** 2021-07-01

**Authors:** Kacuk Cikal Nugroho, Ubaidillah Ubaidillah, Retna Arilasita, Margono Margono, Bambang Hari Priyambodo, Budi Purnama, Saiful Amri Mazlan, Seung-Bok Choi

**Affiliations:** 1Mechanical Engineering, Sekolah Tinggi Teknologi Warga Surakarta, Sukoharjo 57552, Indonesia; nugroho.k.c.89@gmail.com (K.C.N.); arghaa849@gmail.com (M.M.); bambang.hari.priyambodo@gmail.com (B.H.P.); 2Mechanical Engineering, Universitas Sebelas Maret, Surakarta 57126, Indonesia; 3Department of Physics, Universitas Sebelas Maret, Surakarta 57126, Indonesia; arilasita@gmail.com (R.A.); bpurnama@mipa.uns.ac.id (B.P.); 4Malaysia Japan International Institute of Technology, Universiti Teknologi Malaysia, Kuala Lumpur 54100, Malaysia; amri.kl@utm.my; 5Department of Mechanical Engineering, The State University of New York, Korea (SUNY Korea), Incheon 21985, Korea

**Keywords:** magnetorheological fluid (MRF), nano-additives, strontium cobalt ferrite, particle sedimentation

## Abstract

This study investigated the effect of adding strontium (Sr)-doped cobalt ferrite (CoFe_2_O_4_) nanoparticles in carbonyl iron particle (CIP)-based magnetorheological fluids (MRFs). Sr-CoFe_2_O_4_ nanoparticles were fabricated at different particle sizes using co-precipitation at calcination temperatures of 300 and 400 °C. Field emission scanning electron microscopy (FESEM) was used to evaluate the morphology of the Sr-CoFe_2_O_4_ nanoparticles, which were found to be spherical. The average grain sizes were 71–91 nm and 118–157 nm for nanoparticles that had been calcinated at 300 and 400 °C, respectively. As such, higher calcination temperatures were found to produce larger-sized Sr-CoFe_2_O_4_ nanoparticles. To investigate the rheological effects that Sr-CoFe_2_O_4_ nanoparticles have on CIP-based MRF, three MRF samples were prepared: (1) CIP-based MRF without nanoparticle additives (CIP-based MRF), (2) CIP-based MRF with Sr-CoFe_2_O_4_ nanoparticles calcinated at 300 °C (MRF CIP+Sr-CoFe_2_O_4_-T300), and (3) CIP-based MRF with Sr-CoFe_2_O_4_ nanoparticles calcinated at 400 °C (MRF CIP+Sr-CoFe_2_O_4_-T400). The rheological properties of these MRF samples were then observed at room temperature using a rheometer with a parallel plate at a gap of 1 mm. Dispersion stability tests were also performed to determine the sedimentation ratio of the three CIP-based MRF samples.

## 1. Introduction

Magnetorheological fluid (MRF) is a smart material that contains micron-sized magnetic particles dispersed in silicone oil, a non-magnetic carrier fluid [1,2]. Although MRFs has a liquid-like structure in their normal state, they can transform to a solid-like structure in milliseconds when an external magnetic field is introduced and revert to their liquid-like structure when the external magnetic field is removed [3]. This is because the magnetic field induces a dipole moment in the MRF particles which forms a chain-like structure. This chain-like structure provides resistance that increases the shear stress and viscosity of MRF [4,5,6]. This characteristic makes MRF ideal for use in semi-active shock absorbers [7,8], seismic dampers [9], brakes [10], prosthetic legs [11], clutches [12], and haptic sensors [13,14].

Multiple magnetic materials, such as iron (II, III) oxide (Fe_3_O_4_) [15,16], calcium iron oxide (CaFe_2_O_4_) [17], manganese ferrite (MnFe_2_O_4_) [18], cobalt ferrite (CoFe_2_O_4_) [19], and magnesium ferrite (MgFe_2_O_4_) [20], have been studied for use in MRF. However, of these materials, carbonyl iron particles (CIP) draw considerable attention due to their soft magnetic characteristics and high magnetic saturation [21,22]. However, the high density of CIP causes a serious problem in the stability of MRF dispersion. As such, the sedimentation stability of MRF can be increased by either modifying the shape of the particles [23], coating the particles with surfactants [24], or introducing nanoparticles as an additive [25]. Of these three methods, introducing nanoparticles as additives has been found to be the most effective and efficient way of increasing the sedimentation stability of MRF [26].

The use of hard magnetic materials as nano-additives in MRF has received considerable attention in recent times [2,27,28,29]. One such material is cobalt ferrite (CoFe_2_O_4_), a hard magnetic material with good oxidation stability, high Currie points, many magnetostrictive properties, and high mechanical properties [30,31]. The introduction of CoFe_2_O_4_ nano-additives has been found to not only increase the sediment stability of MRF, but strengthen shear stress as well [32]. The hard magnetic properties and magnetic saturation of CoFe_2_O_4_ can be further improved by doping it with strontium (Sr) [33].

As evident from the above literature survey, so far, the introduction of Sr-CoFe_2_O_4_ nano-additives in MRF has never been studied. Consequently, the technical contribution of this work is to investigate the field-dependent rheological property and sedimentation stability of MRF samples containing Sr-CoFe_2_O_4_ nanoparticles synthesized using co-precipitation at calcination temperatures of 300 and 400 °C to obtain materials with different particle sizes. This enabled us to observe the effect of introducing varying sizes of Sr-CoFe_2_O_4_ nanoparticles in CIP-based MRF to storage modulus and sedimentation ratio, and hence compare with the MRF sample without nanoparticle additives.

## 2. Materials and Methods

### 2.1. Preparation of Sr-CoFe_2_O_4_ Nanoparticles

Sr-CoFe_2_O_4_ nanoparticles were synthesized using co-precipitation as previously reported by Arilasita et al., 2018 [33]. The precursor was prepared using cobaltous nitrate hexahydrate (Co(NO_3_)_2_·6H_2_O) (Merck), strontium(II) nitrate tetrahydrate (Sr(No_3_)_2_·4H_2_O), iron(III) nitrate nonahydrate (Fe(NO_3_)_3_·9H_2_O) (Merck, Darmstadt, Germany), and sodium hydroxide (NaOH) (Merck, Darmstadt, Germany). A stoichiometric amount of 0.0009 M Co(NO_3_)_2_·6H_2_O (Merck, Darmstadt, Germany), 0.002 M Fe(NO_3_)_3_·9H_2_O (Merck, Darmstadt, Germany), and 0.0001 M Sr(NO_3_)_2_·4H_2_O was then dissolved in distilled water until homogenous. 4.8 M of NaOH was then dripped in, one drop at a time, to the obtained solution to begin the titration process. During titration, the solution was stirred at 100 rpm and maintained at 95 °C. Once the solution had cooled to room temperature, the precipitate was washed with distilled water until a clean precipitate was obtained. The cleaned product was then dried in an oven at 95 °C for eight hours to remove residual water. Samples that were free from residual water were calcined at 2 variations of temperature (300 °C and 400 °C) with 10 °C/min of temperature increase speed, 5 h of holding time and cooled in the furnace until reaching room temperature. In this study, two variations of nanoparticles were then obtained: (1) Sr-CoFe_2_O_4_ calcinated at 300 °C (Sr-CoFe_2_O_4_-T300) and (2) Sr-CoFe_2_O_4_ calcinated at 400 °C (Sr-CoFe_2_O_4_-T400). The whole synthesis process is shown in Figure 1. 

### 2.2. Preparation of CIP-Based MRF

Grade CC carbonyl iron particles (CIP) from Badische Anilin und Soda Fabrik (BASF) were used as the primary MRF base. Silicon oil with a dynamic viscosity of 0.5 Pa.s was used as a carrier fluid. Three CIP-based MRF samples were then prepared: (1) CIP-based MRF without nanoparticle additives (CIP-based MRF), (2) CIP-based MRF with Sr-CoFe_2_O_4_ nanoparticles calcinated at 300 °C (MRF CIP+Sr-CoFe_2_O_4_-T300), and (3) CIP-based MRF with Sr-CoFe_2_O_4_ nanoparticles calcinated at 400 °C (MRF CIP+Sr-CoFe_2_O_4_-T400). The composition of each sample is shown in Table 1.

### 2.3. Characterisation

The X-ray diffraction (XRD) test was performed to obtain crystal structures of the Sr-CoFe_2_O_4_ nanoparticles that had been calcinated at 300 °C and 400 °C. The morphology of nanoparticles was observed using a field emission scanning electron microscope (FESEM) from JEOL, JSM-7800F, Tokyo, Japan. Before the tests were carried out, all types of MRF samples were stirred for 3 min to maintain homogeneity. The magnetic properties of the Sr-CoFe_2_O_4_ nanoparticles and CIP-based MRF samples were measured using a vibrating sample magnetometer (VSM) MicroSense, FCM-10, Lowell, MA, USA. The rheological properties of the CIP-based MRF samples were obtained using a rotational parallel-plate rheometer (Modular Compact Rheometer (MCR) 302™, Anton Paar^®^, Graz, Austria) with a PP-20 plate and a temperature control device at 25 °C. The data collection procedure on rheology testing was carried out for 5 s at each data collection point. The gap between the upper plate and the lower plate was set at 1 mm. The dispersion stability of the CIP-based MRF samples was obtained by measuring the sedimentation ratio of the samples at room temperature for 1800 min.

## 3. Results and Discussion

The XRD pattern of Sr-CoFe_2_O_4_ nanoparticles was shown in Figure 2. The spectral peak of XRD pattern has good compatibility with International Centre for Diffraction Data card (ICDD) number 221086. It shows that Sr-CoFe_2_O_4_ nanoparticles have an inverse-spinel shape with FCC atomic arrangement and space group Fd 3m. The particle properties can be observed at the highest peak characteristics in XRD pattern. The crystal size of the nanoparticles was determined by the Scherrer’s equation (Equation (1)) [34].
(1)$$D=\frac{0.9\lambda }{\beta \cos\theta }$$
where λ is the wavelength of the X-ray radiation. θ is the Bragg’s angle, and β is the full width at half maximum (FWHM) of the peak. The lattice constant (a) of nanoparticle Sr-CoFe_2_O_4_ was calculated using Equation (2). Where h, k, and l are Miller indices [35].
(2)$$a=\frac{\lambda }{2\sin\theta }\;\sqrt{h^{2}+k^{2}+l^{2}}$$
The density of crystallite (ρ) was obtained using Equation (3) [36].
(3)$$\rho =\frac{Z.M}{N_{A}.a^{3}}$$
where Z is the number of molecules per unit cell, $N_{A}$ is Avogadro’s number, and M is the weight of molecule. The calculation of the given parameters is shown in Table 2.

The size and morphology of the Sr-CoFe_2_O_4_ nanoparticles that were examined using a FESEM are shown in Figure 3. The particle sizes were obtained by measuring the diameter of the particles using the ImageJ image processing program at 100 random particle points. Most of the particles in the Sr-CoFe_2_O_4_-T300 and Sr-CoFe_2_O_4_-T400 samples were found to be 71 nm to 91 nm and 108 nm to 136 nm, respectively. Therefore, increasing the calcination temperature increased the size of the nanoparticles [37,38,39].

The hysteresis curve of the Sr-CoFe_2_O_4_ nanoparticles and CIP-based MRF samples are shown in Figure 4. Figure 4a shows that the Sr-CoFe_2_O_4_-T400 sample had a higher magnetic saturation than the Sr-Fe2O4-T300 sample at 39.29 emu/g and 35.77 emu/g, respectively. As such, larger particle sizes had higher magnetic saturation [37,40]. As seen in Figure 4b, the CIP-based MRF had high magnetic saturation (127.35 emu/g). The addition of Sr-CoFe_2_O_4_-T300 or Sr-CoFe_2_O_4_-T400 nanoparticles were found to increase the magnetic saturation of the CIP-based MRF [25,41,42]. More specifically, the magnetic saturation of the CIP-based MRF with Sr-CoFe_2_O_4_-T400 (MRF CIP+Sr-CoFe_2_O_4_-T400) was higher than in the CIP-based MRF with Sr-CoFe_2_O_4_-T300 (MRF CIP+Sr-CoFe_2_O_4_-T300). The magnetic saturation of the MRF CIP+Sr-CoFe_2_O_4_-T300 and MRF CIP+Sr-CoFe_2_O_4_-T400 samples were 134.86 emu/g and 146.70 emu/g, respectively. The magnetic properties of nanoparticles and MRF samples were shown in Table 3.

A rotational rheometer, in controlled shear rate mode, was used to measure the properties of the CIP-based MRF. This test was conducted at a shear rate of 0.1 s^−1^ to 2000 s^−1^ in magnetic flux densities of 0 T, 0.19 T, 0.31 T, and 0.43 T. Figure 5 shows the curves of the CIP-based MRF samples with shear stress as a function of shear rate at various magnetic flux densities. The closed and open symbols represent the CIP-based MRF and the CIP-based MRF with Sr-CoFe_2_O_4_ nanoparticles, respectively. These symbols were used to represent the same samples in Figure 6 as well. As seen in Figure 5, the shear stress increased as the magnetic field increased. This enhanced shear stress was a consequence of the particle chain structures from dipole–dipole interactions between the MRF particles [23].

Figure 5a compares the flow curves of the shear stresses of the CIP-based MRF and the MRF CIP+Sr-CoFe_2_O_4_-T300, while Figure 5b compares the flow curves of the shear stresses of the CIP-based MRF and the MRF CIP+Sr-CoFe_2_O_4_-T400. The curve shows that the MRF CIP+Sr-CoFe_2_O_4_-T300 and T400 samples had higher shear stress values than the CIP-based MRF sample at every magnetic flux density. This indicated that the addition of Sr-CoFe_2_O_4_ nanoparticles at both annealing temperatures improved the shear stress of CIP-based MRF. This was because Sr-CoFe_2_O_4_ nanoparticles filled the gaps between the CIP and increased the contact area between the particles [32]. Shear stress has an important role in MRF performance. The higher shear stress values indicate that the MRF needs a weaker magnetic field to control its MR characteristics. Decreasing the intensity of the magnetic field means that less electrical energy is needed. 

Figure 5c shows the relationship curve between the shear stress and the shear rate of all MRF samples without a magnetic field. The curve shows that when the shear rate was small, the shear stress value was close to zero. When shear rate was increased, the shear stress increased linearly. This phenomenon shows that all MRF samples without a magnetic field tend to behave like Newtonian fluids [16]. However, when a magnetic field was applied, the MRF had a steady shear stress value in each shear rate range and behaved like a Bingham fluid [43].

The viscosity of the three CIP-based MRF samples as a function of shear rate at different magnetic flux densities is shown in Figure 6. In the absence of a magnetic field, the viscosity along the shear rate in all MRF samples shows a relatively constant value and behaves like a Newtonian fluid. However, when a magnetic field is applied, the curve shows that the viscosity decreased as the shear rate increased and behaved like shear tinning [44]. On the other hand, the viscosity of the MRF showed an obvious increase as the strength of the magnetic field increased. This increase in viscosity plays an important role in controlling MRF characteristics and is as important as shear stress characteristics.

The yield stress of the MRF sample was obtained by analyzing the shear stress curve against shear rate using the Bingham plastic equation as bellow [45]:(4)$$\tau =\tau_{y}+p\dot{\gamma };\;\dot{\gamma }\gg \;$$
where τ is the shear stress, $\tau_{y}$ is the yield stress, p is the plastic viscosity, and $\dot{\gamma }$ is shear rate. The yield stress curve against the magnetic field of all MRF samples can be seen in Figure 7. The MRF CIP+Sr-CoFe_2_O_4_-T400 sample had the highest yield stress at all magnetic fields followed by the MRF CIP+Sr-CoFe_2_O_4_-T300 sample then the CIP-based MRF sample. The MRF CIP+Sr-CoFe_2_O_4_-T400 sample had higher yield stress than the MRF CIP+Sr-CoFe_2_O_4_-T300 sample as the nanoparticles of the MRF CIP+Sr-CoFe_2_O_4_-T400 sample had a higher magnetic saturation. As such, the attractive forces between the particles of the Sr-CoFe_2_O_4_-T400 sample were greater than in the Sr-CoFe_2_O_4_-T300 sample. This higher attractive force between particles strengthens the chain structure of MRF particles thereby increasing the yield stress [46].

Figure 8 shows the correlation between storage modulus and loss modulus against shear strain during oscillatory testing at a constant frequency of 6.28 rad/s. The storage modulus of all three MRF samples significantly increased after the magnetic field was applied. In addition, the storage modulus in all MRF samples increases with the strengthening of the magnetic field. This was because the magnetic field induces the particles and causes an attractive force between the MRF particles. At low shear strains, the storage modulus tends to be stable at certain values. This region where the storage modulus value is stable is called the linear viscoelastic (LVE) region [45,47]. In the LVE region, the curve of storage modulus against shear strain tends to be flat. Similar results were also obtained in other studies [48,49,50,51]. However, at certain shear strain values, the storage modulus decreased significantly. When the storage modulus begins to fall, the value of the shear strain is called critical strain [52]. The critical strain of all three CIP-based MRF samples increased as the magnetic fields increased. The magnetic field induces MRF particles and causes attractive forces between the particles. The greater the particle attraction force, the greater the critical strain required to break the particle chain bonds [48,50]. The critical strain of each MRF sample is presented in Table 4.

The loss modulus of all MRF samples at small shear strain has a relatively low value. As the shear strain increases, the loss modulus also increases. However, when the shear strain was in the critical strain region, the loss modulus tends to peak and decrease after that [48,53]. At a small shear strain value, all MRF samples have a storage modulus value greater than the loss modulus. Up to a certain shear strain value, the storage modulus curve intersects with the loss modulus curve. After that point, the value of the loss modulus on the MRF is greater than the storage modulus. This phenomenon shows that, at low shear strain, MRF has properties that resemble a solid-state. At a certain shear strain, the MRF properties will change to be more like a liquid-state [54,55].

The dispersion stability test was performed at room temperature for 1800 min. The sedimentation ratio was determined using the following equation.
(5)$$\mathrm{SR}=\frac{H_{\mathrm{MRF}}}{H_{S}}$$
where SR is sedimentation ratio, H_MRF_ is the height of the MRF, and H_S_ is the height of the sedimentation. The correlation between sedimentation ratio and time is shown in Figure 9. The MRF CIP+Sr-CoFe_2_O_4_-T300 sample had the lowest reduction in sedimentation ratio followed by MRF CIP+Sr-CoFe_2_O_4_-T400 sample then the CIP-based MRF. After 1800 min, the sedimentation ratios of the CIP-based MRF, MRF CIP+Sr-CoFe_2_O_4_-T300m, and MRF CIP+Sr-CoFe_2_O_4_-T400 samples were 19.7%, 21.1%, and 19.9%, respectively. As such, the MRF CIP+Sr-CoFe_2_O_4_-T300 sample had the highest dispersion stability followed by the MRF CIP+Sr-CoFe_2_O_4_-T400 sample then the CIP-based MRF. This phenomenon indicated that the addition of nanoparticles increased the dispersion stability of the MRF. This was most likely because Brownian motion of the nanoparticles caused collisions between the CIP particles and the nanoparticles thereby preventing the MRF particles from settling [56]. Additionally, the MRF CIP+Sr-CoFe_2_O_4_-T300 sample had higher dispersion stability than the MRF CIP+Sr-CoFe_2_O_4_-T400 sample. This was because the particles of the Sr-CoFe_2_O_4_-T300 sample, which were smaller, have a higher Brownian motion than the particles of the Sr-CoFe_2_O_4_-T400 sample [26]. 

Table 5 presents previous research on adding nanoparticles to MRF. Previous research has shown that the addition of nanoparticles can improve the dispersion stability of the MRF. The effectiveness of nano-additives on the dispersion stability was calculated by the equation bellow
(6)$$\mathrm{effectiveness}=\frac{{\mathrm{SR}}_{\mathrm{MRF}\;\mathrm{with}\;\mathrm{nanoadditives}}}{{\mathrm{SR}}_{\mathrm{MRF}\;\mathrm{without}\;\mathrm{nanoadditives}}}\;100\%$$
when compared with other studies, the effectiveness of the nano-additive in this study was still relatively small (107%). Many things affect the dispersion stability of MRF, these are the size of the nano-additive particles [57], the mass concentration of the dispersed particles [29], and the ambient temperature in the MRF operation [58]. 

## 4. Conclusions

The Sr-CoFe_2_O_4_ nanoparticles used in this study were prepared via co-precipitation and calcination at 300 and 400 °C. Sr-CoFe_2_O_4_-T400 nanoparticles had larger particle sizes and magnetization saturation than Sr-CoFe_2_O_4_-T300 nanoparticles. The addition of Sr-CoFe_2_O_4_ nanoparticles to the CIP-based MRF was found to increase its magnetic saturation. The MRF CIP+Sr-CoFe_2_O_4_-T400 sample had the highest magnetic saturation, followed by the MRF CIP+Sr-CoFe_2_O_4_-T300 sample, then the CIP-based MRF sample. Moreover, the addition of Sr-CoFe_2_O_4_ nanoparticles to CIP-based MRF increased its shear stress. The MRF CIP+Sr-CoFe_2_O_4_-T400 sample had higher shear stresses than the MRF CIP+Sr-CoFe_2_O_4_-T300 sample. This was because the nanoparticles of the Sr-CoFe_2_O_4_-T400 sample had higher magnetic saturation. As such, these nanoparticles have a stronger dipole moment resulting in a stronger particle chain structure. While the MRF CIP+Sr-CoFe_2_O_4_-T400 sample had better shear stress characteristics, the MRF CIP+Sr-CoFe_2_O_4_-T300 sample had better dispersion stability. As such, the dispersion stability of all three samples was (1) the MRF CIP+Sr-CoFe_2_O_4_-T300 sample, (2) the MRF CIP+Sr-CoFe_2_O_4_-T400 sample, and (3) the CIP-based MRF sample in descending order.

Research on the addition of nanoparticles has been shown to strengthen shear stress and dispersion stability in MRF. However, various studies have shown that there is an optimum concentration of nanoparticles that can be added to MRF. At certain concentrations, the addition of nanoparticles can reduce the shear stress of MRF [2,60]. Therefore, it is necessary to conduct further research on the optimum concentration of Sr-CoFe_2_O_4_ nanoparticles which can increase MRF properties, and it is necessary to study whether the size of the nanoparticles will affect the optimum concentration of nanoparticles.

## Figures and Tables

**Figure 1 materials-14-03684-f001:**
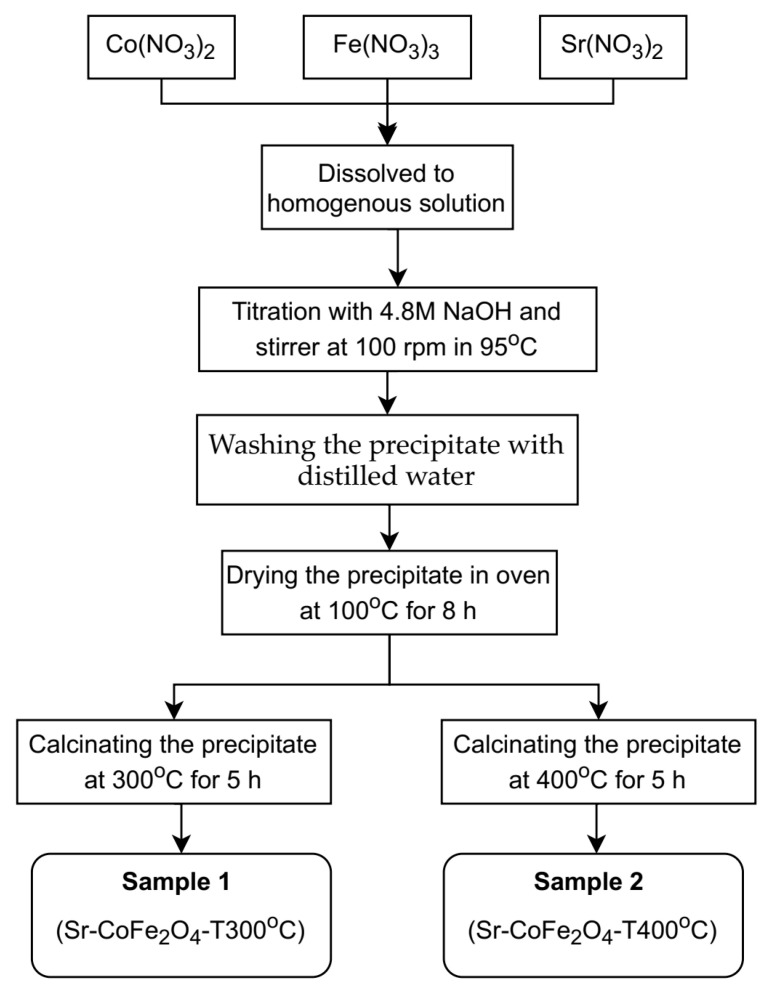
Diagram process of synthesizing nanoparticles Sr-Co Ferrite.

**Figure 2 materials-14-03684-f002:**
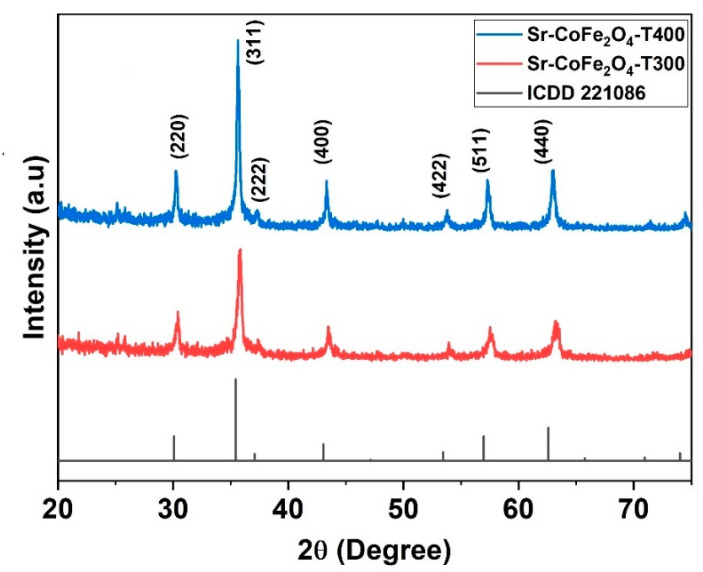
XRD pattern of nanoparticles Sr-CoFe_2_O_4_ with difference calcination temperature.

**Figure 3 materials-14-03684-f003:**
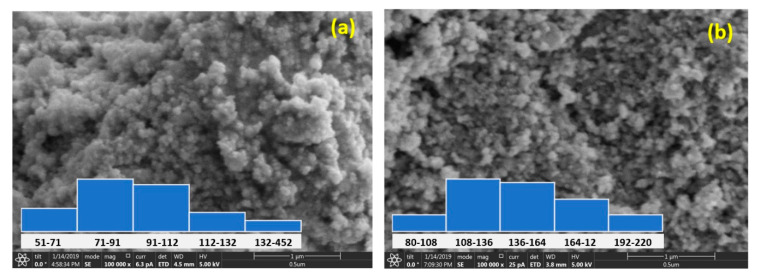
FESEM of Sr-CoFe_2_O_4_ nanoparticles (**a**) with calcination temperature 300 °C (**b**) with calcination temperature 400 °C.

**Figure 4 materials-14-03684-f004:**
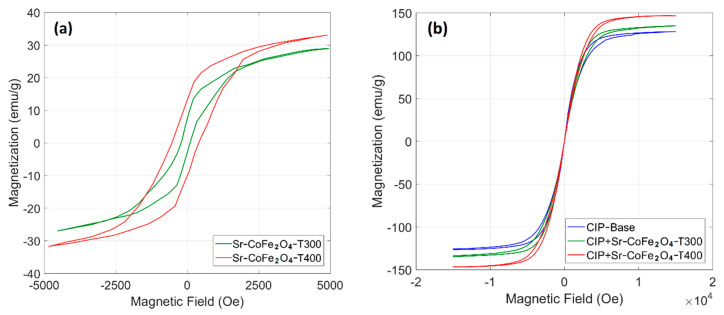
VSM test of (**a**) nanoparticles and (**b**) MRF samples.

**Figure 5 materials-14-03684-f005:**
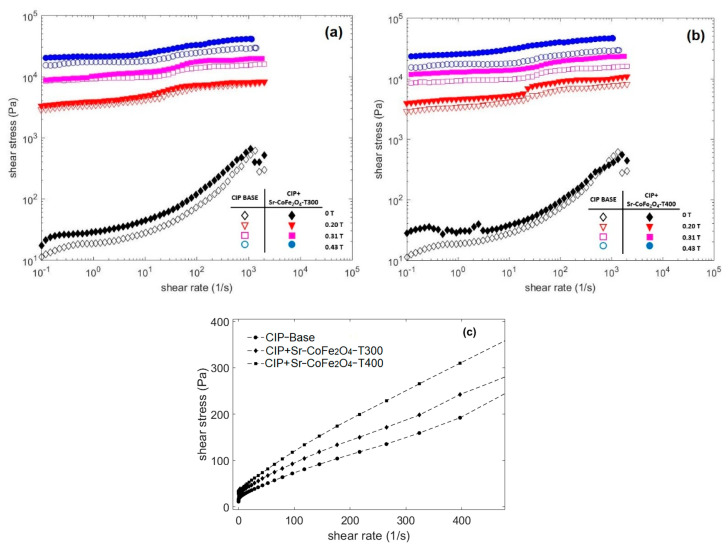
Shear stress of MRF as a function of shear rate at different magnetic fields (**a**) Comparison between MRF CIP-base and MRF CIP+Sr-CoFe_2_O_4_-T300 (**b**) Comparison between MRF CIP-base and MRF CIP+Sr-CoFe_2_O_4_-T400 (**c**) Comparison between MRF CIP-base, MRF CIP+Sr-CoFe_2_O_4_-T300 and MRF CIP+Sr-CoFe_2_O_4_-T400 at 0T in linear scale.

**Figure 6 materials-14-03684-f006:**
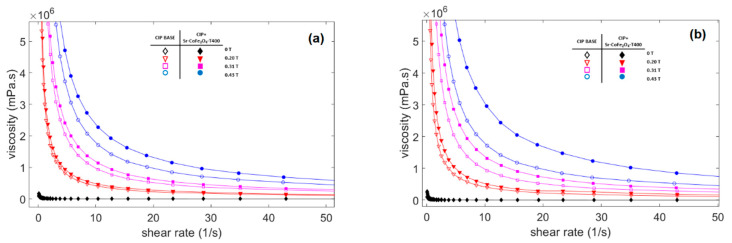
Viscosity curve of MRF samples against shear rate at different magnetic fields (**a**) Comparison between MRF CIP-base and MRF CIP+Sr-CoFe_2_O_4_-T300 (**b**) Comparison between MRF CIP-base and MRF CIP+Sr-CoFe_2_O_4_-T400.

**Figure 7 materials-14-03684-f007:**
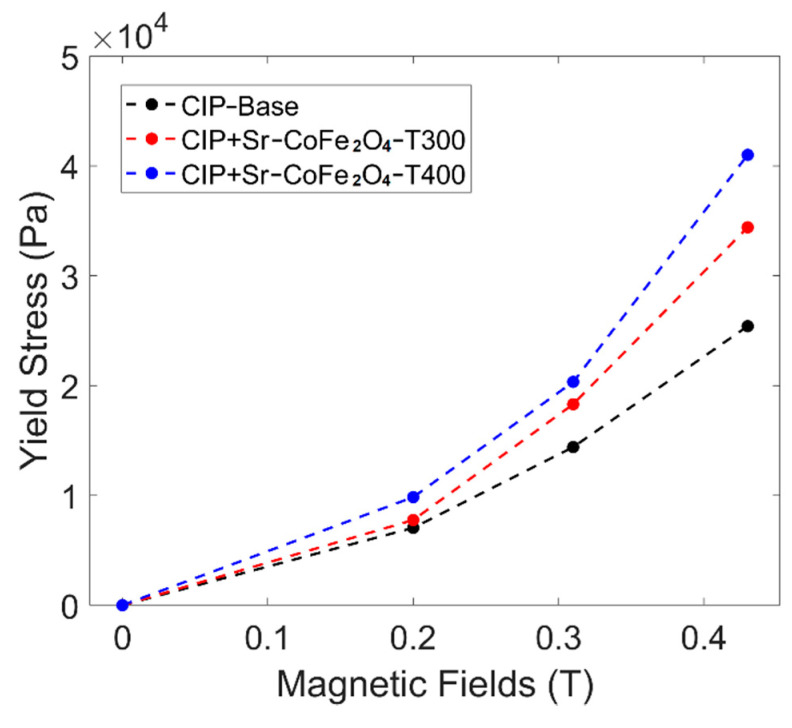
Shear stress curve against magnetic fields of all MRF samples.

**Figure 8 materials-14-03684-f008:**
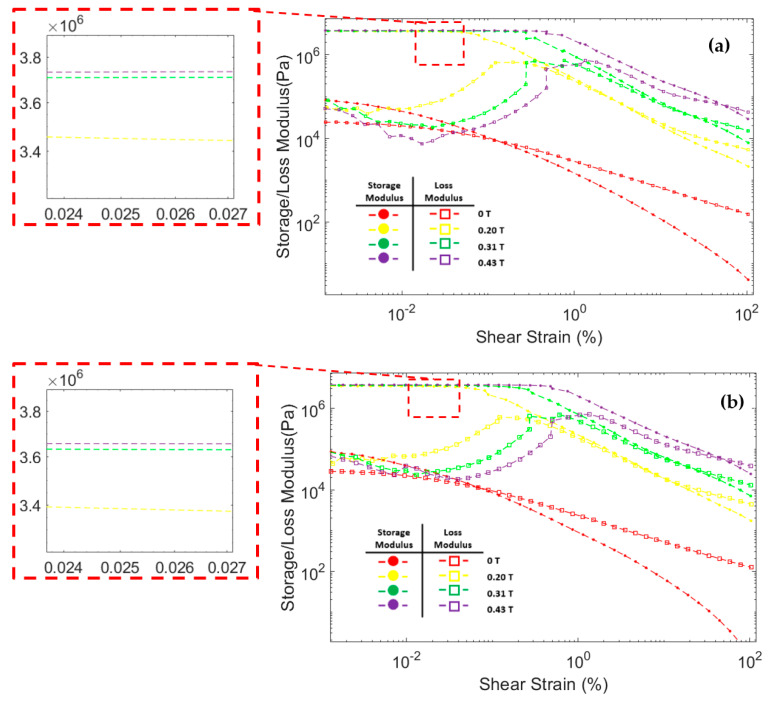
Storage modulus and loss modulus curve against the shear strain of MRF (**a**) CIP-Base (**b**) CIP+Sr-CoFe_2_O_4_-T300 (**c**) CIP+Sr-CoFe_2_O_4_-T400.

**Figure 9 materials-14-03684-f009:**
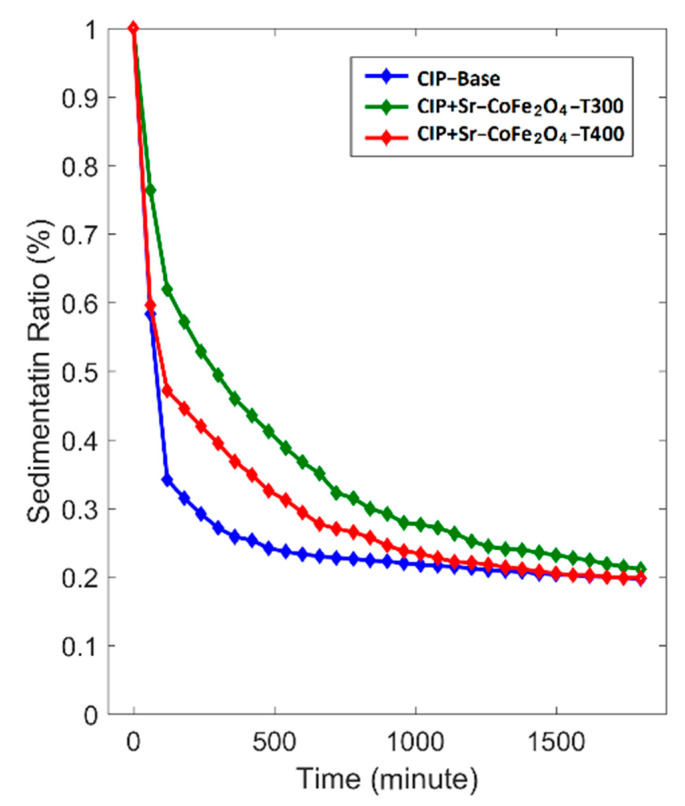
Sedimentation ratio curve against time of MRF samples.

**Table 1 materials-14-03684-t001:** The composition of MRF Samples.

Sample	%wt CIP	Additives	
		Nanoparticle	%wt
MRF CIP-base	70	-	-
MRF CIP+Sr-CoFe_2_O_4_-T300	70	Sr-CoFe_2_O_4_-T300	1
MRF CIP+ Sr-CoFe_2_O_4_-T400	70	Sr-CoFe_2_O_4_-T400	1

**Table 2 materials-14-03684-t002:** Crystallite size, lattice parameter and density of nanoparticles Sr-CoFe_2_O_4_ at difference calcination temperature.

Calcination Temperature (°C)	Crystallite Size, D (nm)	Lattice Parameter, a (Å)	Density, ρ (g/cm^3^)
300	17.26 ± 0.08	8.315 ± 0.015	5.49 ± 0.030
400	21.56 ± 0.11	8.344 ± 0.001	5.43 ± 0.002

**Table 3 materials-14-03684-t003:** Magnetic properties of nanoparticles and MRF samples.

Sample	Magnetic Saturation (emu/g)	Coercivity (Oe)
Sr-CoFe_2_O_4_-T300	35.77	154.13
Sr-CoFe_2_O_4_-T400	39.25	477.39
MRF CIP-base	127.35	127.35
MRF CIP+Sr-CoFe_2_O_4_-T300	134.86	134.86
MRF CIP+Sr-CoFe_2_O_4_-T400	146.70	146.70

**Table 4 materials-14-03684-t004:** The critical strain of MRF samples.

Magnetic Field (T)	Critical Strain (%)		
	CIP-Base	CIP+ Sr-CoFe_2_O_4_-T300	CIP+ Sr-CoFe_2_O_4_-T400
0	0.003	0.004	0.005
0.2	0.059	0.065	0.084
0.31	0.261	0.272	0.358
0.43	0.451	0.486	0.639

**Table 5 materials-14-03684-t005:** Previous research on the stability of dispersions and nano-additives in MRF.

Ref.	Nano Particle	Nano Size	CIP	CIP Size	Carrier Fluids	Sample			Sedimentation Test		
						No.	CIP (wt%)	Nano (wt%)	Time (h)	SR (%)	Effectiveness
[59]	Fe_3_O_4_ hollow spherical shape	200–300 nm	CIP	2 µm	oil	1	50	0	45	39	-
						2	50	0.1	45	58	148%
[60]	Fe_3_O_4_ with cellulose	-	BASF, 7.86 × 10^3^ kg/m^3^	7.2 µm	Silicon oil 3.50 × 10^−4^ m^2^/s)	1	60	0.5	800	60	-
						2	60	1	800	70	116%
						3	60	2	800	80	133%
[50]	Fe_3_O_4_/Sepiolite	d = 50 nm, l = 300–500 nm	BASF, CM Grade, 7.6 g/cm^3^	4 µm	silicone oil, KF-96, 1000 cSt	1	50	0	48	35	-
						2	50	0.1	48	60	171%
[41]	γFe_2_O_3_, coated oleic acid	8.34 nm	Sigma Aldrich, density 7.86 g/mL	10 µm	hydraulic oil (Mobil DTE 25)	1	80.98	0	250	54	-
						2	80.98	1	250	59	109%
[2]	γFe_2_O_3_	9 nm	Sigma Aldrich, density 7.86 g/mL	1–5 µm	hydraulic oil (Mobil DTE 25)	1	80.98	0	48	26	-
						2	75.98	5	48	31	119%
						3	70.98	10	48	35	134%
[25]	ZnFe_2_O_4_	12 nm	CIP	-	silicone oil	1	30	0	7500	38	-
						2	30	1	7500	64	168%
[61]	ZnFe_2_O_4_	150–200 nm	BASF, Grade CC, 7.8 g/cm^3^	2–4 µm	Silicone oil, 1000cSt	1	50	0	36	15	-
						2	50	0.5	36	30	200%
[29]	SrFe_12_O_19_	110 nm	CIP, 4.5 g/cm^3^	1–2.5 µm	Heat transfer oil, 0.92 g/cm^3^	1	30	0	20	45	-
						2	45	0	20	74	164%
						3	60	0	20	83	184%
						5	25	5	20	63	140%
						6	22.5	7.5	20	80	177%
						7	20	10	20	88	195%
[32]	CoFe_2_O_4_	180 nm	BASF, CM Grade, 7.6 g/cm^3^	4 µm	Silicone oil KF-96–100cSt	1	30	0	24	51	-
						2	30	0.1	24	71	139%
[31]	CoFe_2_O_4_	188–222 nm	BASF, CC grades	-	Silicone oil	1	30	0	30	19.7	-
						2	30	1	30	24.9	126%
This Work	Sr-CoFe_2_O_4_, calcinated 400 °C	71–91 nm	BASF, CC grades	-	Silicone oil	1	30	0	30	19.7	-
						2	30	1	30	21.1	107%

## Data Availability

Data sharing is not applicable for this article.

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
