# Peer review of "The Effect of Sr-CoFe_2_O_4_ Nanoparticles with Different Particles Sized as Additives in CIP-Based Magnetorheological Fluid"

_materials, 2021, doi:10.3390/ma14133684_

Round 1

Reviewer 1 Report

The paper „The Effect of Sr-CoFe2O4 Nanoparticles with Different Particles Sized as Additives In CIP-Based Magnetorheological Fluid“ deals with interesting topic and the results, mainly the increase in saturation magnetization of the MR fluid based on CIP particles with addition of Sr-CoFe2O4 nanoparticles are worth publishing. Even though the results are very interesting, before its publication in journal Materals I recommend major revision of the paper. Authors should improve the paper mainly at two points: (i) authors should explain the mechanism behind some observations and (ii) the paper should be thoroughly checked since there are many small mistakes (some of them discussed below).

  • Calcination process should be described more detailed to be able to repeat it for other research groups – atmosphere, heating ramp, if the samples were kept in the furnace to cool down, etc.

  • Addition of only 1 wt% of prepared nanoparticles to the system led to significant increase in Ms values of the MR suspensions (Fig. 4b). Could you please give some explanation behind this huge increment?

  • Line 182 – In my opinion the behavior of the MRFs in the absence of a magnetic field is rather pseudoplastic than Newtonian. Consider please the slope of the curve. Clear pseudoplastic behavior of prepared MRFs can be also seen from Fig. 6 (discussed on the line 192).

  • Line 202 – change „was“ to“ were“

  • Regarding Figure 8, it is stated on lines 221-222 that „The critical strain of all three CIP-based MRF samples increased as the 221 magnetic fields increased“. According to my knowledge, it is common that with increasing magnetic field, the critical strain should decrease due to increasing rigidity of the created structures. Please comment on this.

  • Figure 9. The legend should be corrected. The name of axis-y should be corrected.

Reviewer 2 Report

The paper investigated the influence of Sr-CoFe2O4 Nanoparticles additives on the rheological properties of MR fluids. The paper has been well written, except for some minor grammatical issues that should be revised. The following comments should be addressed before the manuscript be recommended for publication:

  1. Section 2.2, the type of the carrier oil and its properties should be mentioned.
  2. Eqs (1) – (3) should be referred to the references.
  3. The properties of the Mr fluids have been investigated under magnetic fields up to 0.43 T. However, the results presented in Figures 5 and 6 do not show any saturation in the shear stress and viscosity of the MR fluids with respect to the magnetic field. It is necessary to investigate the MR fluids' properties at higher fields to determine the saturation magnetic field above which the properties of the MR fluids do not change.
  4. Figures 5 and 6, the legends should be revised, and the types of materials should be clearly shown in the legends.
  5. The yield stress of the MR fluids at different magnetic fields should be obtained and compared for different fluids.
  6. In addition to the storage modulus that has been presented in Figure 8, the loss modulus of the MR fluids should also be shown and discussed.
  7. Figure 8, the storage modulus of the MR is strangely constant for strains~<0.1%. However, the linear viscoelastic region is a known behavior of MR fluids; such a constant and flat curve has not been shown in the literature. It is suggested to check the results for possible errors.
  8. The testing procedure, such as the time that the MR fluids have been stirred before the tests, and the temperature, and also the data acquisition procedure such as the number of cycles that have been used to obtain the storage modulus should be mentioned.
  9. The values of the critical strain of the MR fluids at different magnetic fields should be mentioned and discussed.
  10. The values of the storage modulus at the different magnetic fields should be mentioned for different MR fluids in LVE region to compare the effect of the additives on the storage modulus. Based on Figure 8, the additives do not affect the storage modulus in the presence of the magnetic field. Why does changing the magnetic field from 0.2 T to 0.42 T not affect the storage modulus in the LVE region?
  11. Frequency is a critical variable that significantly affects the rheological and viscoelastic properties of MR fluids. Variation of the storage and loss moduli with respect to the frequency (especially for strains in the LVE region) should be presented.

Round 2

Reviewer 1 Report

The paper has been significantly improved. It can be now accepted for its publication.